# Estimation of Stride Length, Foot Clearance, and Foot Progression Angle Using UWB Sensors

Ji Su Park [1], Bohyun Lee [1], Shinsuk Park [2,*] and Choong Hyun Kim [1,*]

1   Center for Bionics, Korea Institute of Science and Technology, Seoul 02792, Republic of Korea; gene0219@kist.re.kr (J.S.P.); bh.lee@kist.re.kr (B.L.)
2   Department of Mechanical Engineering, Korea University, Seoul 02841, Republic of Korea
*   Correspondence: drsspark@korea.ac.kr (S.P.); chkim@kist.re.kr (C.H.K.)

**Abstract:** Stride length (SL), foot clearance (FC), and foot progression angle (FPA) are the key parameters for diagnosing gait disorders. This study used the distance data between two feet measured by ultra-wideband (UWB) sensors installed on shoes and proposed a method for estimating the three gait parameters. Here, a method of compensating the offset of the UWB sensor and estimating the distances between a base sensor installed on one foot during the stance phase and three UWB sensors on the other during the swing phase was applied. Foot trajectory was acquired in a gait experiment with ten healthy adults walking on a treadmill. The results were compared with those obtained using a motion capture system (MCS). The UWBs sensor displayed average errors of 45.84 mm, 7.60 mm, and 2.82° for SL, FC, and FPA, respectively, compared with the MCS. A similar accuracy level was achieved in a previous study that used an inertial measurement unit (IMU). Thus, these results suggest that UWB sensors can be extensively applied to sensor systems used to analyze mobile gait systems.

**Keywords:** ultra-wideband; gait parameter; stride length; foot clearance; foot progression angle





## 1. Introduction

Gait analysis assesses and measures the temporal and spatial characteristics of the body during motion. Gait characteristics data can be used as diagnostic parameters for various adult diseases, resulting in numerous clinical researchers focusing on developing technologies for measuring gait characteristics [1,2]. Among gait parameters, stride length (SL), gait speed (GS), and foot clearance (FC) are used to determine the risk of falling [3–5]. In addition, the foot progression angle (FPA) could be applied to examine the characteristics of the load applied to the knee joint while assessing degenerative arthritis [6–9].

Clinicians have reported that the gait asymmetry problem caused by slow GS [10] or low FC [11,12] could be improved through visual biofeedback. In addition, a study that fed the FPA to a subject walking on a treadmill confirmed that the subject improved the FPA on their own [13]. However, these studies were inapplicable in their daily life because they were mostly conducted in experimental environments that used motion capture systems (MCSs). Therefore, numerous studies have analyzed gait parameters using wearable devices. An inertial measurement unit (IMU) is commonly used for gait analysis with wearable devices. The SL [14–16], GS [17,18], gait phase [19–21], joint angle [22,23], and FC [24–26] can be estimated using these devices. The IMU can also be used simultaneously with an ultrasonic sensor to measure the step length and stride width [27].

The disadvantage of using an IMU is the drift phenomenon, which is the occurrence of a cumulative error during speed acquisition by integrating acceleration data. Therefore, researchers conducting mobile gait analysis have improved the accuracy by applying the zero-velocity update (ZUPT) method [28], which corrects the sensor measurement by constraining the speed to zero if the sensor is determined to be at rest owing to the

set threshold value. However, although ZUPT may be effective under circumstances of a short activity time and repeated state of rest, it does not function when: (1) the stance phase only occurs for a very short period, such as while fast walking or running; and (2) the sensor used to detect zero velocity is affected by noise and the drift is not adequately corrected [29–31].

Ultra-wideband (UWB) is an alternative to the IMU with the abovementioned problems. UWB sensors determine distance using the time-of-flight (TOF) between two UWB sensors based on the reception and transmission of radio signals, making it highly advantageous compared with IMU because of its capability to measure distances directly. UWB technology has mainly been used to track locations within indoor spaces using fixed UWB anchor [32,33]. However, as the power consumption of UWB is reduced [34] and the measurement accuracy is increased [35] owing to recent technological developments, the utilization of UWB for gait parameter analysis, such as tracking the joint angle [36,37] and measuring the SL [38], is increasing.

The limitations of previous studies using UWB sensors are as follows: (1) since the signal receiver is installed in a fixed location, such as a wall or tripod, it can only operate within a designated place; and (2) gait analysis is possible only in the double stance phase.

To overcome these limitations, this study proposes a UWB sensor system (Figure 1) that tracks foot trajectory using four UWB sensors installed on each foot to analyze gait characteristics. First, the trilateration method was applied, which involved simultaneously applying the distance data acquired from each UWB sensor to determine the overlapping location and estimate the actual distance between the sensors. However, the distance data provided by the UWB sensors had errors owing to variations in the antenna direction. Therefore, the variation in the strength of the received sensor signals was considered to compensate for this error. The SL, FC, and FPA data for clinical assessments were determined by conducting a gait experiment using the developed UWB sensor system. Simultaneously, an MCS was applied to consider the data as the gold standard and review the measurement accuracy of the UWB sensor system.

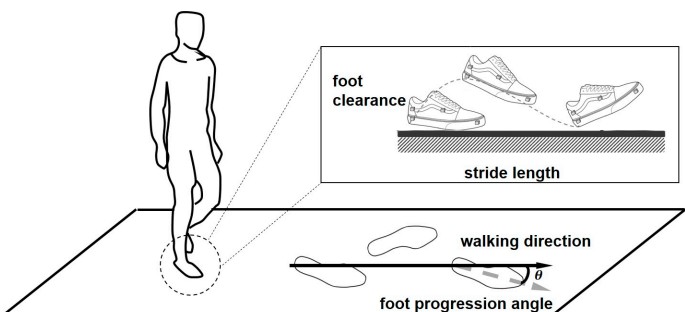

**Figure 1.** Schematic illustration of the UWB sensor system to analyze gait characteristics.

## 2. Materials and Methods

### 2.1. Hardware Description

As shown in Figure 2a, the equipment used in this experiment is a UWB system, comprising UWB sensors attached to each shoe to measure the distance (collected on a personal computer (PC)), an MCS that measured the location data of the reflection marker attached to the shoe, and a device that synchronized the data of the two systems.

The UWB system included a UWB sensor (model: DWM1000, Qorvo, Greensboro, NC, USA) and an Arduino Nano. The UWB sensor was installed on an in-house printed circuit board (PCB) and fabricated to have a total size of 43 mm (width) × 20 mm (length) × 15 mm (height) and a data acquisition speed of 100 Hz. The trilateration method involves attaching three anchors and a tag to a shoe, as shown in Figure 2b, using eight UWB sensors (four in each shoe). The UWB sensors were placed between the two shoes facing each other so that wireless signals could be sent and received without interruption. As illustrated in Figure 2c, the tag on each foot measures the distance between the three

anchors located on the opposite foot. The data were transmitted to and stored on a PC connected to a wire and used in the analysis. Gait data were measured using the MCS (Vicon Nexus MX, Oxford Metric, Oxford, UK) installed in the gait experiment space. The data measured by MCS were considered as the gold standard and were applied to evaluate the trajectory estimated by the UWB system. A reflection marker for motion capture was attached to the PCB for the UWB sensors, as shown in Figure 2a,b. The location of the reflective marker was placed at a site that does not affect the antenna signal of the UWB sensor. Meanwhile, a push switch (5 V) for triggers was attached to the right-foot tag, and its signal was used to synchronize the UWB data obtained from the gait experiment with the motion capture data before comparing them (Figure 2a).

The gait experiment proceeded with the participant walking on a treadmill (M-gait, Motek, Amsterdam, The Netherlands) wearing shoes equipped with UWB sensors and MCS markers.

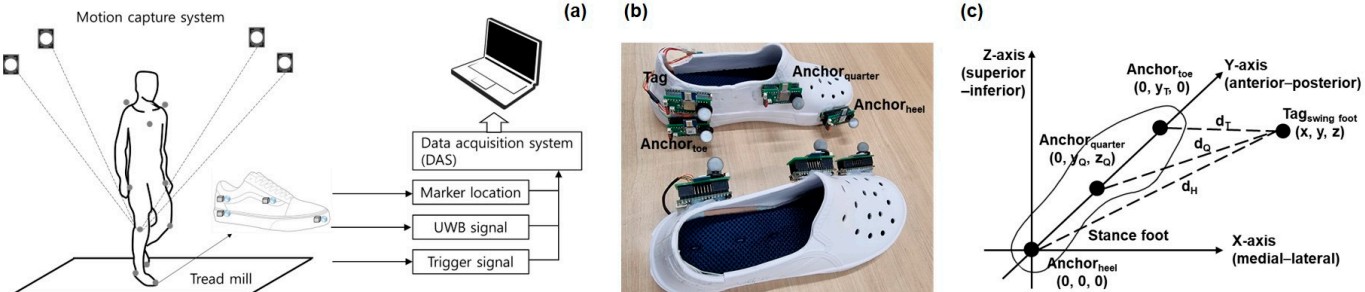

**Figure 2.** Test setup: (**a**) Data acquisition system; (**b**) UWB and MCS marker installation; (**c**) Coordinate system for the UWB sensors.

## 2.2. Participants and Test Method

The participants in the gait experiment were 10 healthy adults (5 males and five females) in their 20 s and 30 s (Table 1). They were devoid of neurological diseases and gait disorders and were capable of normal gait. The experimental protocol was approved by the Institutional Review Board (IRB) of the Korea Institute of Science and Technology (KIST). All participants provided written informed consent for participation in the study.

**Table 1.** Subject information.

| Subject Group | Male | Female |
|---|---|---|
| Number of participants | 5 | 5 |
| Age (years) | $32.2 \pm 3.7$ | $28.0 \pm 4.5$ |
| Height (m) | $1.75 \pm 0.04$ | $1.61 \pm 0.06$ |
| Weight (kg) | $71.0 \pm 6.1$ | $49.6 \pm 6.0$ |

Each participant installed a UWB system and MCS marker on a shoe that fit their size and participated in the experiment. The participants ascended the treadmill and stood waiting while gazing toward the front until they received a cue from the test performer. When they received the signal, they walked at designated speeds (2, 3, and 4 km/h) for 3 min and stopped the gait according to the stopping speed. The conductor first operated the trigger for data synchronization, then started the treadmill at the designated speed, along with the cue sign. When the walking time exceeded three minutes, the conductor stopped the trigger, declared the end of the test to the subjects, and stopped the treadmill. The first five steps after the trigger operation and the final five steps before the treadmill stopped were excluded from the collected gait experiment data during data analysis to increase accuracy.

### 2.3. Calculation of Gait Parameters Using UWB

This section describes the method for estimating gait parameters using a UWB sensor. The gait characteristic analysis comprises three steps, as shown in Figure 3a. The three gait parameters, SL, CL, and FPA (see Figure 3b,c), were used in the analysis.

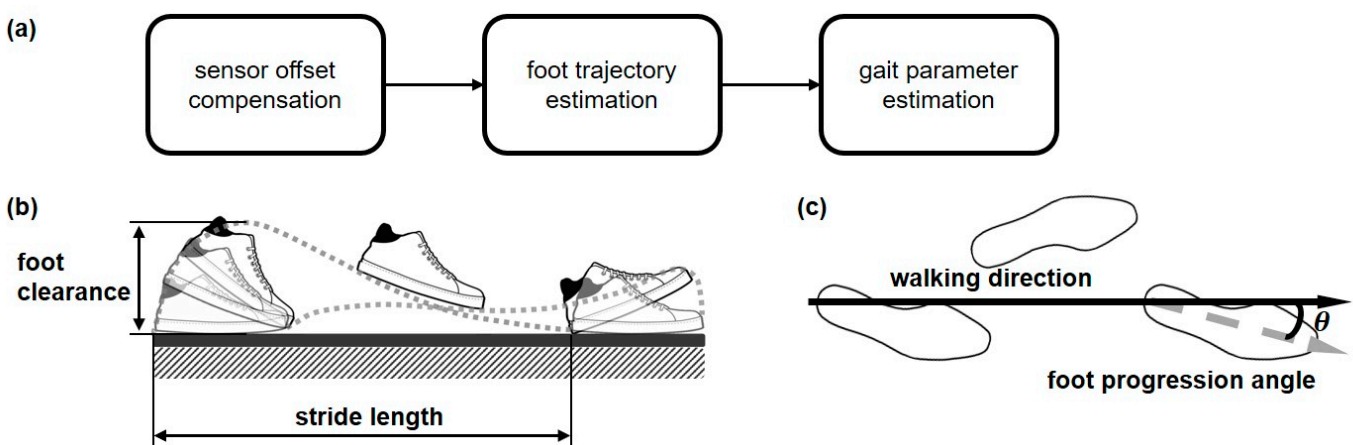

**Figure 3.** Gait analysis and its parameters: (**a**) Flowchart of the gait parameter estimation algorithm; (**b**) Gait parameters: foot clearance and stride length; (**c**) Gait parameter: foot progression angle.

### 2.3.1. UWB Sensor Offset Compensation

The UWB sensor used in this study performs distance measurements using the two-way ranging (TWR) method [39]. In this method, (1) the TOF, as the time required to transmit and receive wireless signals between the two directions of the anchor (as the fixed antenna) and tag (as the mobile antenna) is determined; and (2) the distance between the two sensors is measured. According to the UWB sensor datasheet (model: DWM1000), the precision of the sensor resolution was estimated to be 10 cm. However, the resolution of the oscillator mounted on the UWB was 16 ps [38], and according to the latest research, it has been proven that the UWB sensor can measure more stably at a short distance than the proposed precision. However, this sensor can experience signal delays according to variations in the surrounding temperature and antenna direction. Therefore, most studies to resolve this issue used the offset compensation method provided by Qorvo Inc. (Greensbor, NC, USA) [40].

The method used by previous researchers involved the variation in the distance between the anchor and tag terminals; measurement of the distance, signal strength, and temperature; and generation of an offset compensation function using the signal strength and temperature used to measure the distance. This method may improve the accuracy of the UWB sensor according to offset compensation in a laboratory environment. However, if the compensation method assumes that the anchor and tag face each other, it is infeasible to compensate for the offset according to the variation in the direction of the antenna. In reality, the offset varies according to the variation in the relative angle between the two UWB sensors [41], and the antenna angle varies periodically while walking, which causes the occurrence of a residual error after offset compensation.

This research used support vector regression (SVR) to compensate for the residual error generated by the variation in the antenna angle according to the gait cycle. Therefore, before the gait characteristics experiment, the following procedure was conducted for the distance data-offset compensation.

1. Before the treadmill gait experiment, the participants wore shoes equipped with UWB sensors and walked on the treadmill for 5 min. In this experiment, the signal strength and distance data measured by the UWB sensor according to the gait cycle before offset compensation was measured, and the location of the motion capture marker next to the UWB sensor was acquired.

2. The function that compensates for the offset of the sensor consists of SVR, which is composed of the signal strength and distance data measured using the UWB sensor, and the target variable Y, which is composed of the motion capture marker data. The composed dataset D is expressed by Equation (1). The UWB distance data used in the dataset were obtained after applying a third-order Savitzky–Golay filter [38].

$$(x_i, y_i)_{i=1}^l, \quad x_i \in R^N, \quad y_i \in \{-1, 1\} \tag{1}$$

3. To distinguish different classes using input data, the target variable should be mapped to the domain of a higher dimension, and the hyperplane is expressed as a shape space vector with mapping ψ to the shape space Z. The linear decision function composed of the weight vector and the bias vector pair $(\omega, b)$ is as follows:

$$f(x) = w^T x + b \tag{2}$$

4. Weight and bias can be estimated by minimizing the Euclidean norm. The minimization equation composed of cost function C and slack variable ξ is as follows:

$$\text{Minimize } \frac{1}{2}\|w^2\| + C \sum_{i=1}^{n} (\xi_i + \xi_i^*) \tag{3}$$

$$\text{subject to } y_i(w{\cdot}z_i + b) \geq 1 - \xi_i$$

5. Equation (3) solves this issue using positive Lagrange multipliers. The corresponding equations are as follows:

$$\text{Maximize } W(\alpha) = \sum_{i=1}^{l} \alpha_l - \frac{1}{2} \sum_{i=1}^{l} \sum_{j=1}^{l} \alpha_i \alpha_j y_i y_j z_i {\cdot} z_j \tag{4}$$

$$\text{subject to } \sum_{i=1}^{l} y_i \alpha_i = 0, \ 0 \leq \alpha_i \leq C, \ i = 1, \ 2, \ \cdots, \ l$$

6. Finally, the kernel function was used to map the data space to the shape space. In addition, a linear kernel is used to compensate for the offset of the data acquired from the UWB sensor. Previous research results [42] can be referred to for further details on the SVR.

### 2.3.2. Foot Trajectory Estimation

The UWB sensor, after compensation, was installed on the shoe, as shown in Figure 2. The location coordinates of the tag were estimated using the gait experiment. In general, the location coordinates are estimated from the relative location of the tag, based on fixed anchor coordinates. However, the coordinates of the anchor cannot be fixed in mobile-type smart insoles. Therefore, for each step in this study, the location of the foot tag in the swing phase was calculated based on the anchor on the foot in the stance phase.

To determine the location of the foot, the coordinate system was first defined based on the anchor located on the big toe ($Anchor_{toe}$) and that located at the heel of the foot ($Anchor_{heel}$) in the stance phase. Here, the location coordinates of the tag, defined as $(x, y, z)$, were $Anchor_{heel}$ as $(0, 0, 0)$, and that of the anchor located at the three-fourths point of the shoe length ($Anchor_{quater}$) as $(0, y_Q, z_Q)$, and that of X as $(0, y_T, 0)$. Then, the distance between each anchor and tag can be expressed as follows:

$$d_T{}^2 = x^2 + (y - y_T)^2 + z^2 \tag{5}$$

$$d_Q{}^2 = x^2 + (y - y_Q)^2 + (z - z_Q)^2 \tag{6}$$

$$d_H{}^2 = x^2 + y^2 + z^2 \tag{7}$$

The above equations can be evaluated to find the tag coordinates $(x, y, z)$, which are expressed as follows:

$$y = \frac{d_H{}^2 - d_T{}^2 + y_T{}^2}{2y_T} \tag{8}$$

$$z = \frac{d_H{}^2 - d_Q{}^2 + y_Q{}^2 + z_Q{}^2 - 2yy_Q}{2z_Q} \tag{9}$$

$$x = \pm\sqrt{d_H{}^2 - y^2 - z^2} \tag{10}$$

In Equation (10), the positive and negative signs are obtained when the location coordinates of the right and left feet are estimated according to the reference coordinate.

### 2.3.3. Stride Length Estimation

SL refers to the linear distance between the coordinate at the time of starting the swing and at the ending of swing from the foot trajectory measured from one stride (Figure 3b). This is determined as follows:

$$SL = \sqrt{(x_{S1} - x_{S2})^2 + (y_{S1} - y_{S2})^2 + (z_{S1} - z_{S2})^2} \tag{11}$$

$$(x_{S1}, y_{S1}, z_{S1}) = \text{argmax}(y_i), \quad (x_{S2}, y_{S2}, z_{S2}) = \text{argmin}(y_i), \quad i = 1, 2, 3, \ldots, n$$

### 2.3.4. Foot Clearance Estimation

FC is the value with the largest magnitude among the foot trajectory z data measured from one stride.

$$FC = \text{max}(z_i), \quad i = 1, 2, 3, \ldots, n \tag{12}$$

### 2.3.5. Foot Progression Angle Estimation

The FPA is calculated as the sum of the angle between the foot direction vector and $d_H$ vector and the walking direction vector, which can be obtained from the coordinates of the start and end points of the swing (Figure 4), and $d_H$ vector. This is calculated as follows:

$$\theta_w = \cos^{-1}\left(\frac{ST^{n\,2} + d_H^{n+1\,2} - d_H^{n\,2}}{2ST^n d_H^{n+1}}\right) \tag{13}$$

$$\theta_F = \cos^{-1}\left(\frac{y_T{}^2 + d_H^{n+1\,2} - d_T^{n+1\,2}}{2y_T d_H^{n+1}}\right) \tag{14}$$

$$FPA = \theta_w + \theta_T - \pi \tag{15}$$

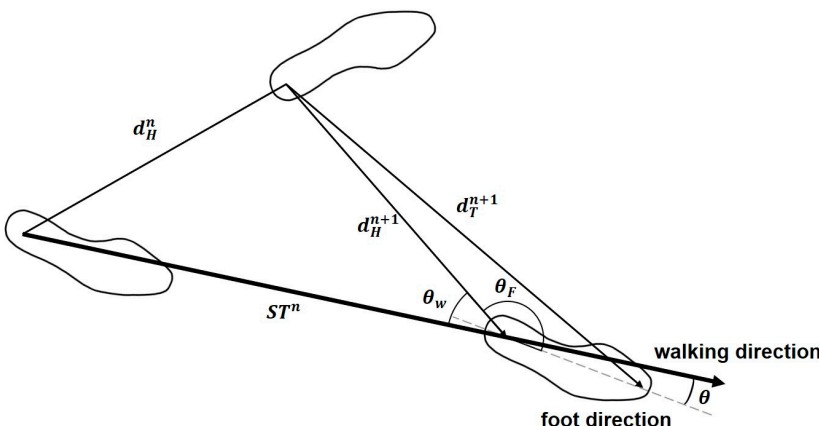

**Figure 4.** Schematic of foot progression angle definition.

## 3. Results

### 3.1. Sensor Offset Compensation

Examples of the motion capture data acquired from the pre-processing experiment to compensate for the UWB sensor data are shown in Figure 5. In Figure 5, the blue line represents the motion capture data used as the gold standard, the black dotted line represents the UWB sensor data, and the red line represents the UWB data that underwent offset compensation using SVR. As shown in Figure 5, compensating for the UWB data using SVR reduces the error. This value was similar to that of the MCS after compensation.

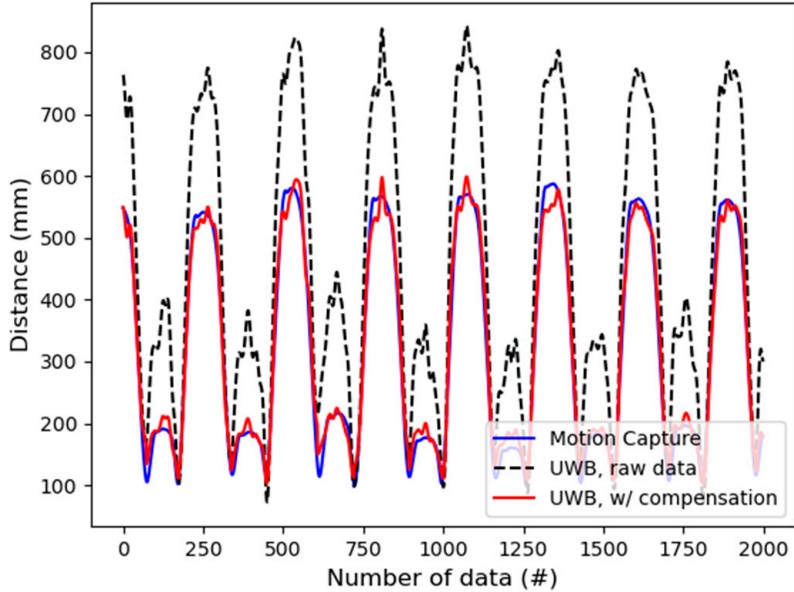

**Figure 5.** Comparison of the UWB data with motion capture system data. Black-dotted line: UWB raw data with the Savitzky–Golay filter, blue-solid line: data obtained using motion capture system, red-solid line: UWB data offset compensated using SVR.

The offset compensation method applied in this research was used to compensate for the offset of the data acquired from six anchors, namely, the $Anchor_{heel}$, $Anchor_{quater}$, and $Anchor_{toe}$ of the right and left foot. These showed mean square errors (MSE) of 10.27, 24.77, 9.27, 8.32, 10.62, and 9.91 mm, respectively. Thus, the average MSE was 12.19 mm.

### 3.2. Results for Gait Parameters

Figure 6 shows the UWB data acquired from participant #3 of the gait experiment and the 3D trajectory of the right foot determined using Equations (8)–(10). The foot trajectory estimated using UWB is shown in Figure 6a. Here, a closed curve is drawn using the combined trajectory of the foot when the tag is in the swing and stance phases. Herein, after the heel strike point, when the foot is in a swing state and higher than the opposite foot, the z-direction location value reverses as the opposite foot lifts from the ground. Therefore, the foot's trajectory at a wholesome swing phase could be illustrated only if the data with the z value of the anchor of the opposite foot is higher than the z value data of the tag location installed on the stepping foot in the stance phase. This is illustrated in Figure 6b.

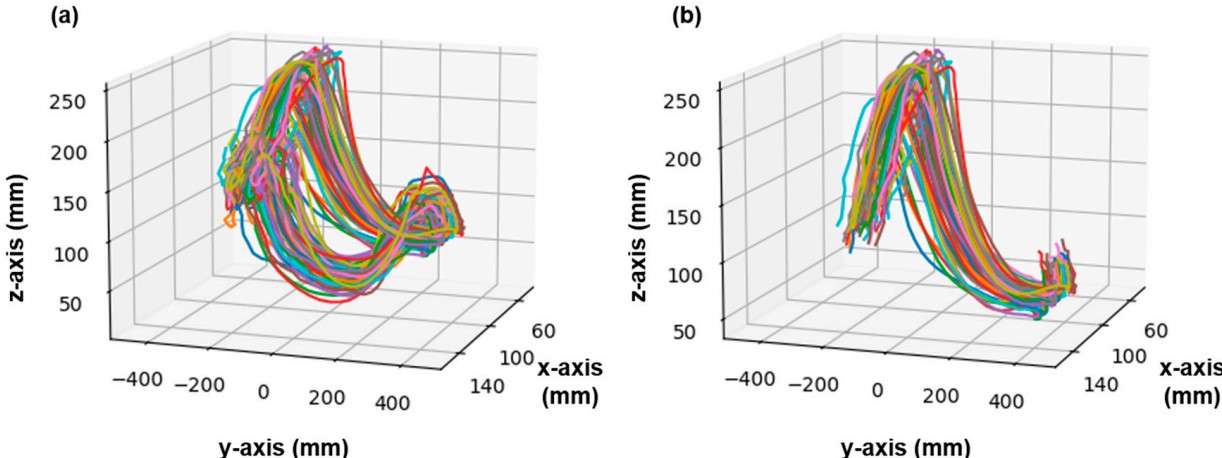

**Figure 6.** Foot trajectory estimation using UWB sensors: (**a**) All the foot trajectory results; (**b**) Only the results estimated in the swing phase.

Figure 7 shows the Bland–Altman plot that assesses SL, FC, and FPA using the 3D trajectory data of the foot during the swing phase. It shows the data acquired from all participants. The x-axis represents the average of the values measured by the MSC (or optical reference) and the UWB system, and the y-axis represents the difference between the two measured values (MCS (or optical reference)—UWB).

It can be seen that the SL data in Figure 7a is divided into two groups, and the average SL values of each group are 719 mm and 874 mm, respectively. The data near SL = 719 mm, were mainly obtained from a group of female participants with heights < 170 cm. The data near SL = 874 mm were mainly from a mixed group of participants with heights ≥ 170 cm. Figure 7 shows that the obtained data were mostly within 95% confidence interval. Therefore, the foot trajectory data estimated using UWB could be considered to be in good agreement with the data measured using MCS.

The data in Figure 7 were analyzed and are presented in Table 2. The differences in SL at speeds of 2, 3, and 4 km/h were 45.13, 46.99, and 45.40 mm, respectively. Furthermore, the average difference in the entire gait speed section was 45.84 mm. The differences in FC at speeds of 2, 3, and 4 km/h were 7.80, 7.45, and 7.54 mm, respectively, and the average difference for the entire gait speed section was 7.60 mm. The differences in the FPA for the same speeds were 2.53°, 2.86°, and 3.12°, respectively, and the average difference for the entire gait speed section was 2.82°. In Figure 8, the average SL, FC, and FPA were measured for each gait speed. Moreover, a 95% confidence interval was identified. Overall, the difference in SL was trivially impacted by the variation in gait speed. In addition, the difference in FC was relatively large when the gait speed was low and tended to converge to a certain value as the gait speed increased. The difference in FPA increased continuously as the gait speed increased, although the amount of variation was marginal.

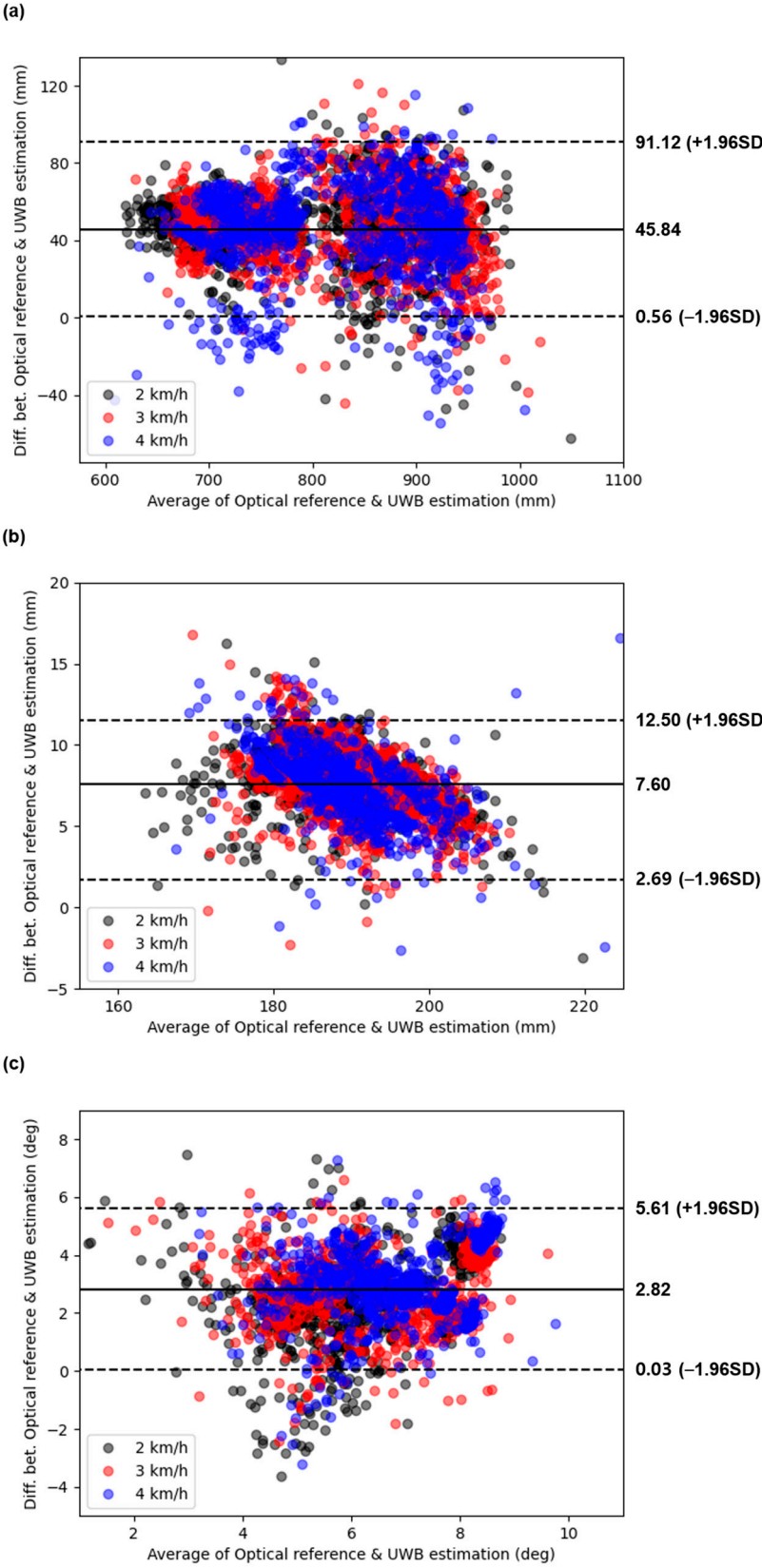

**Figure 7.** Bland–Altman graphs for gait parameter values determined using UWB and motion capture system: (**a**) Stride length; (**b**) Foot clearance; (**c**) Foot progression angle.

**Table 2.** Means and standard deviations of differences in gait parameters between motion capture and UWB approach.

| Subject | Stride Length (mm) | | | Foot Clearance (mm) | | | Foot Progression Angle (°) | | |
|---|---|---|---|---|---|---|---|---|---|
| | 2.0 km/h | 3.0 km/h | 4.0 km/h | 2.0 km/h | 3.0 km/h | 4.0 km/h | 2.0 km/h | 3.0 km/h | 4.0 km/h |
| Sb01 | 40.67 ± 24.61 | 34.63 ± 25.87 | 26.48 ± 46.72 | 8.46 ± 1.84 | 8.17 ± 1.32 | 7.89 ± 1.52 | 1.34 ± 3.26 | 2.10 ± 2.16 | 1.50 ± 1.98 |
| Sb02 | 43.09 ± 23.14 | 47.90 ± 24.78 | 51.82 ± 19.81 | 8.23 ± 1.71 | 7.82 ± 2.11 | 8.63 ± 1.46 | 1.58 ± 1.87 | 1.54 ± 1.01 | 2.95 ± 1.33 |
| Sb03 | 46.48 ± 22.77 | 47.84 ± 25.72 | 50.01 ± 20.36 | 8.22 ± 1.76 | 6.28 ± 1.92 | 6.65 ± 1.92 | 3.12 ± 1.32 | 3.26 ± 1.92 | 4.09 ± 0.95 |
| Sb04 | 43.94 ± 25.26 | 47.13 ± 39.08 | 40.92 ± 13.35 | 8.91 ± 2.06 | 8.41 ± 1.82 | 7.24 ± 1.89 | 2.42 ± 0.24 | 3.31 ± 0.35 | 3.24 ± 0.17 |
| Sb05 | 45.17 ± 33.04 | 47.49 ± 26.57 | 46.14 ± 25.28 | 6.04 ± 2.00 | 7.10 ± 2.77 | 6.45 ± 4.88 | 2.00 ± 0.32 | 2.14 ± 0.41 | 2.19 ± 0.31 |
| Sb06 | 46.10 ± 34.59 | 55.94 ± 23.99 | 52.23 ± 19.17 | 8.68 ± 1.25 | 7.72 ± 0.95 | 7.88 ± 0.94 | 2.68 ± 1.24 | 2.60 ± 0.78 | 3.06 ± 0.89 |
| Sb07 | 44.23 ± 21.50 | 38.96 ± 15.70 | 29.77 ± 43.56 | 5.63 ± 1.45 | 5.74 ± 1.63 | 6.28 ± 1.67 | 3.22 ± 0.21 | 3.14 ± 1.18 | 3.31 ± 1.17 |
| Sb08 | 45.58 ± 10.66 | 48.94 ± 12.61 | 52.47 ± 11.21 | 8.40 ± 0.82 | 8.27 ± 1.30 | 8.51 ± 1.35 | 4.16 ± 0.43 | 4.23 ± 1.42 | 4.70 ± 0.33 |
| Sb09 | 46.68 ± 10.78 | 46.39 ± 18.79 | 44.26 ± 20.32 | 7.82 ± 2.71 | 6.33 ± 3.40 | 7.43 ± 1.09 | 1.80 ± 1.99 | 2.60 ± 1.41 | 2.00 ± 1.14 |
| Sb10 | 47.97 ± 19.05 | 52.13 ± 19.80 | 49.87 ± 8.09 | 8.07 ± 2.01 | 8.78 ± 3.34 | 7.71 ± 2.15 | 2.49 ± 0.54 | 3.17 ± 0.68 | 2.78 ± 0.43 |
| Average | 45.13 ± 21.99 45.84 ± 23.10 | 46.99 ± 22.34 | 45.40 ± 24.83 | 7.80 ± 1.88 7.60 ± 2.50 | 7.45 ± 3.17 | 7.54 ± 2.27 | 2.53 ± 1.54 2.82 ± 1.42 | 2.86 ± 1.33 | 3.12 ± 1.31 |

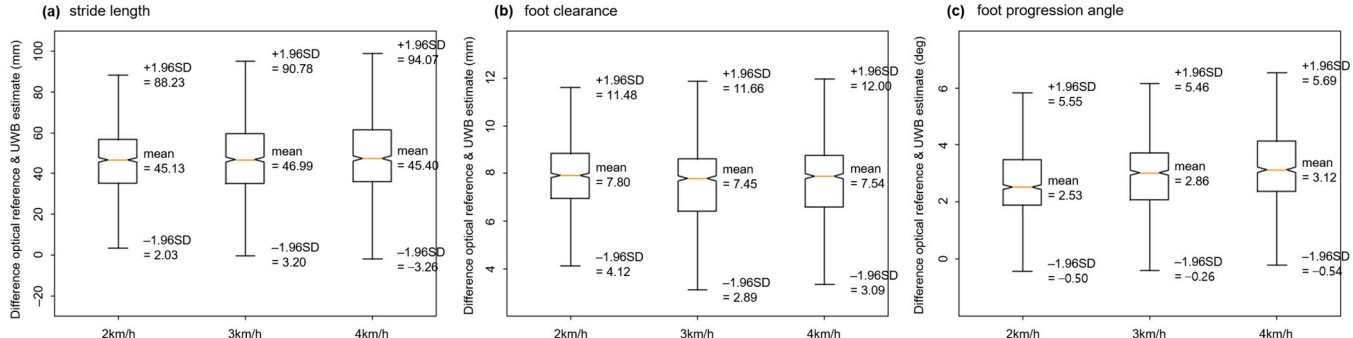

**Figure 8.** Box plot results for difference between motion capture and UWB approach: (**a**) Difference in stride length; (**b**) Difference in foot clearance; (**c**) Difference in foot progression angle test-results.

The estimated gait parameters from UWB and MCS were compared: they show good correlation coefficients ($r^2 > 0.7$) of 0.898 in SL, 0.761 in CL, and 0.743 in FPA.

## 4. Discussion

Table 3 compares the SL, FC, and FPA data obtained in this study (expressed as KIST) with data acquired in previous studies that used an IMU.

**Table 3.** Mean gait parameter difference of motion capture compared with other sensors.

| Gait Parameter | Researcher | Difference of Motion Capture |
|---|---|---|
| Stride length | KIST | 45.84 mm |
| | Uchitomi et al. [43] | 54 mm |
| | Hori et al. [44] | −27 mm |
| Foot clearance | KIST | 7.60 mm |
| | Fan et al. [45] | 4.2 mm (Heel)/13.1 mm (Toe) |
| | Huang et al. [46] | 7.0 mm/2.7 mm (Toe) |
| Foot progression angle | KIST | 2.82° |
| | Wouda et al. [31] | 2.6° |
| | Huang et al. [29] | 2.5° |

The SL estimation error was 45.84 mm in this study, compared to 54 mm [43] and (−) 27 mm [44] in previous studies. However, two previous studies applied IMU sensors to the shank for gait experiments. Therefore, the measurement error for the SL value acquired from this research was considered to be in a range similar to that of the previous research that used an IMU.

FC estimation error was 7.60 mm in this study, compared to 4.2 mm/2.7 mm [45], and 8.6 mm [46] in previous studies. In a study by Fan et al. [45], the FS estimation error varied according to the attachment location of the IMU. The heel clearance and toe clearance measurement errors were 7.0 mm and 2.7 mm, respectively, when the IMU was located at the toes, and 4.2 mm and 13.1 mm, respectively, when located at the heel. This phenomenon was similar to that observed by Huang et al. [46]. These results were attributed to the identical foot experiencing different accelerations according to the part within the three-dimensional movement of the foot. Therefore, the detected foot trajectory was considered to differ according to the installation location of the IMU on the foot. This was also a significant error factor when tracking the foot trajectory using the IMU. In addition, the FC measurement error in this study was within a range similar to that used in previous studies that used an IMU.

The FPA estimation error was 2.82° in this study, compared to 2.6° [31] and 2.5° [29] in previous studies. In a study by Wouda et al. [31], the FPA measurement error was observed to increase to 5.22° when the sensor location was misarranged during the positioning of the coordinate system of the IMU attached to the foot. In summary, the error that occurred during the process of applying ZUPT and arranging the sensor coordinate system in the gait experiment using the IMU sensors was considered to have a significant impact. The measurement error of the FPA acquired from this study differed from the results of previous studies, which used an IMU of approximately 0.22°–0.32°.

Overall, the SL, FC, and FPA values estimated using UWB were similar to those estimated using IMU or displayed marginal differences.

There are two likely sources of error in the methodology with UWB sensors used in this research:

First, in the case of offset removal using SVR, the signal strength information is used to indirectly assess the variation in the relative angle of the anchor and tag attached to different feet. In this case, the relative angle and signal strength of the antenna do not display a linear relationship [40], and the signal strength can vary from the external impact of gadgets, such as Bluetooth devices. Therefore, to address this issue, it is necessary to compensate for the offset more accurately using a continuous gait phase that directly relates the varying antenna angle during walking to the ankle angle.

Second, an error occurred in the hypothesis applied while calculating the foot trajectory using the trilateration method. An anchor with four fixed points is required to determine the foot trajectory using the trilateration method. However, the locations of the two feet in this study were assumed to not intersect in the medial–lateral direction (feet are not entangled), and the number of anchors was reduced to three. In addition, the three locations of the anchor in the stance phase were assumed to be in a fixed coordinate system on the ground before the trajectory of the opposite foot was tracked. However, in reality, the repeated occurrence of a situation wherein one foot is at the heel contact point and the opposite foot is at the heel lift-off point, and the anchor is not entirely fixed to the ground, results in errors in the results of the characteristic analysis.

## 5. Conclusions

This study estimated the three gait parameters SL, FC, and FPA through gait experiments using shoes installed with four UWB sensors on the right and left feet. The accuracy of the results was similar to that of the studies that applied to the IMU.

The UWB sensor used in this study contains measurement error occurrence factors that originate from compensating data or the setting of the coordinate system for identifying the UWB sensor location. However, it enables direct determination of the SL. Furthermore, although it displays a measurement error similar to that in previous studies using IMU sensors, the drift that occurred with the IMU sensors was absent. Therefore, the applicability of the UWB sensor for gait characteristic analysis was considered high.

In future research, based on the results of this study, the gait phase detection method using an insole device could be applied to develop a more accurate foot trajectory tracking technology and conduct gait disorder diagnosis on actual patients.

**Author Contributions:** Conceptualization, J.S.P. and C.H.K.; methodology, J.S.P. and C.H.K.; software, J.S.P.; validation, J.S.P. and C.H.K.; formal analysis, J.S.P. and C.H.K.; investigation, J.S.P. and C.H.K.; data curation, J.S.P. and B.L.; writing—original draft preparation, J.S.P.; writing—review and editing, S.P. and C.H.K.; supervision, C.H.K.; project administration, C.H.K.; funding acquisition, C.H.K. All authors have read and agreed to the published version of the manuscript.

**Funding:** This research received no external funding.

**Institutional Review Board Statement:** The study was conducted in accordance with the Declaration of Helsinki and was approved by the Institutional Review Board of the Korea Institute of Science and Technology (approval no. KIST-HR-001 in 2022).

**Informed Consent Statement:** Informed consent was obtained from all the participants involved in the study.

**Data Availability Statement:** The data presented in this study are available upon request from the corresponding author. The data are not publicly available owing to ethical concerns (because they were obtained in a clinical trial).

**Acknowledgments:** This research was funded by the Korea Institute of Science and Technology (KIST) Institutional Program (project nos. 2E31642 and 2E32341).

**Conflicts of Interest:** The authors declare no conflict of interest.

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
