# Peer review of "Estimation of Stride Length, Foot Clearance, and Foot Progression Angle Using UWB Sensors"

_applsci, doi:10.3390/app13084801_

Round 1

Reviewer 1 Report

Dear authors,

Thank you for your work on detemining gait parameters using UWB.

In my opinion this is a interesting alternative to other ambulatory measurement systems such as based on IMUs.

However, the manuscript can be improved in the following aspects:

- The introduction is not fully covering the scope in which your work is relevant. And is missing citations to works that use distance measurements for gait parameter determination, such as: Weenk 2015 (Ambulatory Estimation of Relative Foot Positions by Fusing Ultrasound and Inertial Sensor Data) and related works.

- ZUPT is stated as being inaccurate for long-term movements, which is not valid. ZUPT is valid for any step, as it only requires a stationary foot for a small period of time to stabilize integration drift from IMUs. This approach is highly succesful in many IMU approaches for gait parameter estimation and there are limitations, but the duration is not typically an issue unless your feet are never in contact with the ground.

- The introduction should be checked for English grammar/vocabulary, as some sentences are unclear.

- The methods mention a strandard deviation for your female size of 0.6m, which is likely a typo.

- The FPA determination is unclear, your measurements are on a treadmill so your walking direction is clearly defined while in real ambulatory settings this is an unknown. So how is this taken into account in your method?

- Your results indicate a bias in your SL/FC/FPA. Is then your method for correcting the UWB bias sufficient? It should be discussed in more detail.

- Your discussion mentions various different abbreviations for FPA, while this should the only one to be used.

- The discussion is lacking comments on clinical relevance, as a FC error of multiple centimeters is likely not clinically relevant, because this can mean the difference between falling or not. While this is less the case for the SL.

- The discussion also doesn't get into limitations of UWB and how this approach is not in all situations optimal, which should be highlighted to help put this approach in the right perspective.

- In general, there can be a more innovative approach to using UWB data that also takes into account signal quality as the angle between tag and anchor has an impact on measurement reliability. Can you then not reweigh your measurements using a kalman filter approach?

Author Response

Thank you very much for your meticulous analysis and suggestions.
Please refer to the attached document.

Reviewer 2 Report

1.      In the abstract section, quantitative data must be included.

2.      Given the "take-home" message at the end of the abstract, the present form was insufficient.

3.      Keywords should have been reorganized alphabetically.

4.      Abbreviation as a keyword is not recommended and encouraged to be changed to become a stand for its abbreviation.

5.      Based on Jamari et al., gait is consist of acting load , range of motion, and cycle. The authors needs to include the valuable comments. Also, relevant reference needs to adopted as follows: Adopted Walking Condition for Computational Simulation Approach on Bearing of Hip Joint Prosthesis: Review over the Past 30 Years. Heliyon 2022, 8, e12050. https://doi.org/10.1016/j.heliyon.2022.e12050

6.      What is the novel bought by the authors in the current submission? Its works have been widely discussed in the past. Nothing something really new in the present form. The lack of a novel seems to make the present submission like to replication/modified work. The authors need to detail their novelty in the introduction section. It is a major concern for rejecting this paper.

7.      Previous research has to be explained in the introduction section, including their work, novelty, and limits, to illustrate the research gaps that will be filled in the current study.

8.      In the last paragraph of the introduction, please explain the objective of the present article.

9.      Recommended to add an additional figure in the introduction section to improve the presentation of the present article.

10.   To make the reader comprehend the workflow of the current study, the authors could include extra examples in the form of figures in the materials and methods rather than merely the dominating text as a present form.

11.   Please explain more clearly the basis of patient selection since the present form was insufficient. Is any standard, procedure, or protocol used? The involved patient is also very small and heterogeneous without real group control. It is urgent since impacting the obtained results would lead to biased analysis. Fatal flaws that need to be addressed after revision.

12.   It also is needed to include more information on tools, such as the manufacturer, the country, and the specification.

13.   Important information that must be mentioned in the publication relates to the error and tolerance of the experimental equipment utilized in this investigation. As a result of the disparate findings in subsequent research by other researchers, it would be a useful discussion.

14.   A comparative assessment with similar previous research is required.

15.   Overall, the discussion in the present article is extremely poor. The Authors must extend their discussion and make a comprehensive explanation.

16.   The conclusion section needs to explain further research.

17.   The authors sometimes reduced a paragraph to just one or two phrases across the whole article, which made the explanation difficult to follow. To make a more thorough paragraph, the writers should expand upon their explanation. It is advised to include at least three sentences in a paragraph, one of which should serve as the primary idea and the others as supporting details.

18.   Five years back literature should be enriched into the reference. MDPI reference is strongly recommended.

19.   I order to reduce the self-citation level, the authors recommend not use their previous work as a reference over.

20.   Due to grammatical problems and linguistic style, the authors should proofread the work.

21.   It is suggested to the authors for providing graphical abstract in the system after revision.

Author Response

(The authors gave the same response as above.)

Round 2

Reviewer 1 Report

Dear authors,

Thank you for revising the manuscript. 

The English has been improved considerably, this makes the work a lot better understandable.

There are a couple of things that require improvement:

- A reflection on the impact of your work, what do these outcomes potentially mean for actual applications? Currently it is just stated that it will be done in future studies, but it does not show the impact of this study.

- A reflection on potential improvements, e.g., what other information can be used to improve the work, like use contact detection? Also the fact that only spatial parameters are estimated and no statements are made on temporal parameters indicate a mismatch between what a clinical gait analysis should entail and what this system can provide.

- ZUPT is not a real problem in the applications you are mentioning, as running is not really possible with the UWB setup, so that argument for a gap in existing works is not adequate. In general, it is not very clear what research gap this work is filling.

- Abstract, line 19, states drift characteristics of IMUs, however, the sensor are not the reason for drift, it is the fact that you integrate the measurement signals, which contain inaccuracies (noise, bias, etc), that all result in a mismatch with the actual positions than those estimated by integration.

- Introduction, line 36, states ankle angle as being the same as FPA, however, the FPA is impacted not only by the ankle, but also by the knee and hip.

- Introduction, line 40, mentions gyroscope seperate from IMU, while an IMU is typically a combination of gyroscope and accelerometer (and sometimes magnetometer). 

- Methods, line 115, informed consent was changed to consent, while informed should be there.

- No statements on how the reflective markers impact the UWB measurements as this will introduce multi-path.

- Table 3 only shows related work errors, the calculated errors for your work should be included in there to quickly understand differences.

Author Response

We appreciate your efforts to review this manuscript.
Excuse us for the late reply to your questions, as our team members recently attended a conference.

Please refer to the attached answers to your questions.

Reviewer 2 Report

Following comments is given in the stage:

1.      Incorporated the studies that adopting gait for their investigation, while it consists of stance phase and swing phase. The recent update literature is suggested to refer as follows: Polycrystalline Diamond as a Potential Material for the Hard-on-Hard Bearing of Total Hip Prosthesis: Von Mises Stress Analysis. Biomedicines 2023, 11, 951. https://doi.org/10.3390/biomedicines11030951

2.      To enhance the present article explanation, additional illustration in introduction section would helpful to easier the understanding.

3.      Basis of statistical analysis needs to explain in more.

4.      The measurement is very affecting to patient involved. One of the characteristics is body mass index. Authors invited to include the explanation in perspective of body mass index. Incorporated literature is needed to: https://jurnaltribologi.mytribos.org/v33/JT-33-31-38.pdf

Author Response

(The authors gave the same response as above.)
